# Is the Environmental Kuznets Curve Still Valid: A Perspective of Wicked Problems

**Jingling Chen [1],\* , Tao Eric Hu [2] and Rob van Tulder [3]**

[1]  Department of Business, Yangzhou University, Yangzhou 225000, China
[2]  Department of Accounting & Information Systems, California State University, Northridge, CA 91330-8372, USA
[3]  Partnerships Resource Center, Erasmus University Rotterdam, Burgemeester Oudlaan 50, 3062 PA Rotterdam, The Netherlands
\*  Correspondence: chenjl@yzu.edu.cn

**Abstract:** Historically, academia has paid much attention to environmental Kuznets curve (EKC) associated hypotheses, and the EKC per se has triggered conflicting reactions since first posited. Yet, all controversies seem not to have any base framework to address further pollution-related strategies. Built upon an extensive critical review of the extant EKC literature, this paper attempts to address the gap by introducing the theory of wicked problems that can be used to reframe the extant EKC research. Integrating and synthesizing the theories and empirical findings of the extant EKC literature, this paper develops a conceptual framework (a research agenda), and suggests that, given humans' bound rationality and societal uncertainties, the EKC pattern may not be valid for the situations of more wicked pollution. Mainly focusing on this type of pollution, the paper contributes to the EKC study in proposing a set of causal relationships built upon the attitudes of societal sectors. The paper points to the necessity of distinguishing the less wicked pollution situations from the more wicked ones that require different practical and academic strategies to deal with. The former can be addressed along with economic growth, and the latter requires proactive attitudes, proactive leadership, and strong organization of societal sectors. In doing so, we hope to advance the conversation surrounding EKC studies and the abatement practice adaption. Contributions of this study and future research avenues for empirical verifications of the theory are then discussed.

**Keywords:** environmental Kuznets curve; wicked problems; societal sector; collaboration; triggering event

## 1. Introduction

Before the advent of the environmental Kuznets curve (EKC), the relationship between economic growth and environmental protection attracted considerable attention dating back to the 1960s. Having fought hard with the main societal problem, namely poverty, the national governments of most developing and underdeveloped countries believed that economic growth provided sustainable resources and technology to tackle environmental problems. Meanwhile, environmental activists and global institutes have kept warning that fast economic growth produces even more industrial and domestic waste, and the accumulation of such waste leads to environmental deterioration and eventually unaffordable prices for the economy and fatal threats to the survival of mankind.

In the 1980s, attention turned to so-called 'sustainable development', which was originally advocated at the United Nations (UN) conference, and referred to the national welfare notion to meet the needs of the present generation without compromising the needs of future generations. Among the rapidly expanding theoretical and empirical literature, EKC associated hypotheses have gained heated

interest in academia. Panayotou (1993) finds a strong link between pollution emissions and economic growth in an inverted-U curve [1]. This was the very first report to introduce the term EKC due to its resemblance to the Kuznets hypothesis of income inequalities, showing that, initially, pollution increases in parallel to income growth, then the line goes down after climbing over a turning point (Figure 1). To elaborate, the economy initially starts from the point of "too poor to be green" [2], then the economy grows at the cost of the environment because "certain environmental problems are linked with the deficiency of economic development"[3], but in the end economic growth sweeps away the previous pollution, resulting in the situation of "grow up first, clean up later"[4].

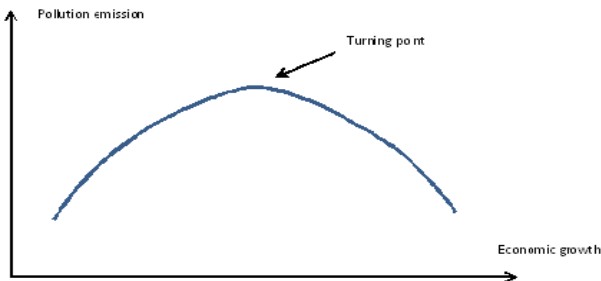

**Figure 1.** Environmental Kuznets Curve.

From the perspective of economic activities, several influential models have been developed for further justification and elaboration. For example, the industry-upgrading model proposes the transformation mode of economic development from the polluting industrial economy to the clean service economy [1,5]. The Green Solow model claims the process of emission reduction arising from exogenous technology progress, leading to a greater usage efficiency in energy and materials that successively reduces the impositions on the natural resources and environment [1]. Further, the displacement model introduces a theory that higher environmental standards in developed areas push or outsource dirty industries to underdeveloped areas—the point that has been proposed in the pollution haven hypothesis [6].

However, EKC associated hypotheses raise the set of multi-faceted questions with the progress of research, especially when the cause of pollution extends to the social system and national context. For instance, Kaika and Zervas (2011) find that the EKC places too much emphasis on the production, arguing that the improvement of technological structure would be offset when the final consumption holds the pollution intensive [7], especially under the constant growth of the volume of urban wastes generated [8]. Further, several other studies [9,10] point out that most empirical studies have assessed the turning point of the EKC based on the average income level of the countries, assuming that the world income is normally distributed—yet the world distribution is indeed highly skewed [11,12], thus it is rather unrealistic to turn to the average income in order to estimate the turning point of the EKC. Furthermore, it has been found that socially deprived areas are exposed to even more environmental degradation [13]. Grimes and Roberts (1997) state that the EKC is valid only for those developed countries with a colonial history, geopolitical powers, and trade superiority [14], therefore the pattern may not be as valid to the less developed world of today [15], implying that the EKC pattern may not be repeatable for developing countries.

As research proceeds, scholars may find it not so optimistic that, as EKC believes, people can deal with pollution well when the economy grows further. Sinha (2010) introduces the case of the USA where more than one million polluted properties are still present in the soil—even after several billion dollars have been invested to deal with it. Sinha therefore warns that some pollutants are just too complicated and costly to remedy, probably impossible to reverse [16]. Thus, the "grow now, clean up later" strategy has been developed and implemented on a very limited knowledge base of toxicological risks and environmental degrading [17]. Meanwhile, it may only be applicable to local pollutants like

urban wastes and water pollution. When it comes to pollution with transboundary impacts, especially global pollutants such as $CO_2$, no country has sufficient incentive to regulate these emissions [18,19].

Furthermore, the methodology and econometric techniques of the extant EKC research have been deeply questioned due to the empirical results that are so controversial for research assumptions and parametric specifications. It is clear that most environmental issues involve multiple interacting systems, replete with not only rational choices but social and institutional uncertainties. One case is the difficulty of action in addressing greenhouse gas emission. In the case, while some nations, organizations, and company frontrunners have taken practical actions in reducing $CO_2$ emission, others feel reluctant to assume the environmental responsibilities and, controversially, think about the relationship between human activities and global climate change not from a science mind but from the perspective of political intentions [20]. The coordinative complexity and different understanding of aims and means among many decision-makers contribute to the huge gap typically arising between ambitious promises and practical performance in grand multi-level programs. Consequently, in the exploration of environmental issues, such problems have been widely recognized from the perspective of the wicked problems theory. Methodologically, the wicked problems cannot be resolved through the traditional analysis of vast amounts of data or conventional statistical analyses [21].

To advance EKC research, we provide a first step towards initiating, organizing, and developing a productive exchange between research on the EKC and wicked problems. From the perspective of wicked problems, this paper seeks to contribute to the literature as it advances a conceptual base framework to help dynamically describe and understand potential pollution–growth relationships and the conditions. The paper is structured as follows. First, the wicked problems theory is introduced to reframe the extant EKC into an attitudinal mindset towards pollution. Through tracing the wickedness sources from the lens of stakeholders' mindsets, the paper proceeds to look at the situation of the more wicked pollution, illustrating that the EKC pattern is not valid for the special situation. Through an extensive critical literature, we propose a set of relationships between such pollution and economic growth, and our viewpoints of critical thinking about the attitudinal transitions of societal actors in dealing with such pollution. Further, we put forward possible effective ways to attain both economic and environmental goals. The contributions of this study and future research avenues for empirical verifications of the theory are also discussed.

## 2. Materials and Methods

### 2.1. Reframing the Extant EKC into an Attitudinal Mind-set towards Pollution

The lens of wicked problems differs from the technical rationality of econometric techniques in many ways. The greatest difference, which makes the perspective so special, lies in that wicked problems are interpreted as a framework of competing values rather than viewing the problems as knowledge gaps [22]. From this perspective, dealing with the associated problems involves a series of collective actions that require attention to multiple factors of the economic- environmental system [23]. As such, the application of econometric techniques in the EKC test is in question because the empirical results of these studies are very sensitive to the assumptions, specifications, and functional forms, very little attention has been paid to omit variable bias and to model adequacy [18,24]. The wicked problems methodologically cannot be resolved through the traditional analysis of vast amounts of data or conventional statistical analyses. Methodologically, the wicked problems cannot be resolved through the traditional analysis of vast amounts of data or conventional statistical analyses [21].

The more societal and the less technical a challenge is, the greater its potential to become wicked. Both starting as an urban planning scientist, Horst Rittel and Malvin Webber found that most societal issues cannot be solved by planning. "As we seek to improve the effectiveness of actions in pursuit of valued outcomes, as system boundaries get stretched, and as we become more sophisticated about the complex working of open societal systems, it becomes ever more difficult to make the planning idea operational" [25].

In 1973, Rittel and Webber co-published a paper which characterizes the societal-focus problems as wicked problems [25]. Following the seminal work, many others have argued that wicked problems should not be treated as a simple or "tame" problem, which is solvable since it can be unambiguously defined, and the approaches and principles for desired outcomes are known and rather transparent [26,27]. Neither can wicked problems be treated as complex problems which the solution to, and the cause–effect relations of, are unclear, but can be known over time after other ways of thinking. They go beyond being too complex to address, even resist the analytical definition because each problem appears to be a symptom of other problems and the cause-effect relations of and solutions to them are largely unclear and unstable. Furthermore, since there is no analytical way to structure, understand, and define the problem, it is impossible to know when it can be solved satisfactorily, at best it is only resolved over and over again [25]. Thus, wicked problems are often described as "social messes" [28] that defy resolution due to enormous interdependencies, uncertainties, and even circularities [26]. Besides the environmental issues, most social problems can be viewed as wicked problems, including poverty, terrorism, discrimination, healthcare and natural source management—they are often intertwined, and one can be the cause of the others.

Moreover, pollution has been fairly described as a "super wicked problem" due to its exacerbating nature and the lack of governance mechanisms [29,30]. On the one hand, its aggravating nature illustrates that time is surely not costless—the longer it takes to address, the harder it will be to do. As pollutants increase, exponentially larger and greater technological advances should be achieved to bring pollutant concentrations down to the desired level. The efforts invested in technological advances are made to make up for the lost time, so as to make it much harder to accomplish the necessary technological innovation. On the other hand, to most of the transboundary pollution, there are no existing institutional frameworks for developing, implementing, and maintaining the laws necessary to address the issues of tremendous spatial and temporal scope. Especially for cases of global pollution such as greenhouse gas emissions, there are no global lawmaking institutions with jurisdictional and legal authorities to address the problem [31]. Thus, the "super wicked" notion speaks to the wicked nature of pollution, in that there is no authority in charge, but it surely needs urgent unified actions.

However, not all wicked problems are equally intractable, and not all pollution problems are equally wicked either. Pollution of lower wickedness is relatively easy to deal with, which can be categorized as tame or complex problems, can be addressed by, such as, the breakthrough of advanced technology, or the establishment and effective enforcement of laws in regulating stakeholders' behaviors. The EKC transition is found very true for the pollutants that have local and regional dimensions, not so wicked, which can be reduced at the relatively low cost of economic growth [32]. When the wickedness goes up to a higher level however, the environment–economy trajectory could go beyond the inverted U speculated in the EKC hypotheses. The more wicked the nature is, the more uncertain the relationships will be.

To depict the possible relationships, we need understand where the wickedness comes from and how the sources determine the pollution emissions. Next, we trace the wickedness sources from rational accounts and social accounts, illustrating that more wicked pollution prevents stakeholders from making rational decisions for bounded rationality, and at the same time, every stakeholder needs to continually optimize their response to the actions of others.

*2.2. Source of Pollution Wickedness from the Lens of Stakeholders' Mind-sets*

2.2.1. Pollution Wickedness of Rational Accounts

According to the theory of bounded rationality, further light can be shed on the wickedness source by three sources of ambiguity—the primary one is knowledge ambiguity. The knowledge base of pollution requires a considerable amount of basic data and sophisticated information, while relevant information is always incomplete, and part of them can be hidden, disguised or intangible. This ambiguity is partly related to the limited technology today, leading to the sole focus of the EKC studies

on selected air and water pollutants, without taking into account toxic and carcinogenic compounds, complicated pollutants in soil and underground water, biodiversity, and desertification, among others. The source of ambiguity is also related to the stakeholders' knowledge-framing, in which too much or too little attention is paid on the relevant information. The attitudes towards pollution may be determined by the process of evidence creation, the selection of evidence, and the interpretation of evidence. The human knowledge frame tends to be pessimistic [33], and human beings are often inclined to focus on dangers, and to be concerned more about what needs to be done now, and less about what can be done later. In either case, an environmental disaster may receive more attention than the climate warming. Additionally, ambiguity can also exist in the political sense as illustrated above. Since the pollution problem is a consequence of population and human activities, the disagreement between stakeholders surrounding pollution often reflects the different emphases they place on various causal factors in different contexts on account of the interests vested therein, thus some information and evidence can be "made" or selected or covered intentionally. From above, we can see there is a big gap can be detected between the "truth"of pollution and stakeholders' perceptions about pollution.

Next, derived from the first is predictive ambiguity. One cannot build predictions on the lack of proper understanding of a phenomenon, nor can one extrapolate development under highly uncertain and unstable conditions. In the technological sector, people even cannot predict it is good news or bad because a breakthrough of technological innovation can bring uncertain risks on human and the environment. For instance, the introduction of genetic technology in food industry. In the political sector, policies in attempt to reducing pollution are not excluded from "unintended consequences". The more urgent a pollution problem is, the higher the likelihood that actions will be taken without thorough planning and design. Even under a careful design, the effects of policies may be based on many assumptions. However, such generative and dynamic complexities of pollution are shaped by, and further fed into, societal, communicative, and structural complexity dimensions in an unpredictable way. Unanticipated and unintended consequences of purposeful actions can be positive, negative, or "perverse" [34]. Here is a demonstrative case. In China, the restriction of odd or even licensed vehicles in smog weathers unintentionally encouraged the rise of private cars in Chinese families. In the global sector, since there are no global lawmaking institutions with jurisdictional legal authorities, policies are more oriented towards local interests. However, it needs to point out that some of these unanticipated consequences may well be deleterious, others might create unforeseen opportunity to stakeholders.

Further derived from the above is intervention ambiguity. Pollution issues are more than their technical complexities, and the societal complexity dimension determines the effectiveness of the chosen interventions. The divergence of interests, values, and power bases reflects the fragmenting motions within the system, and hence a coordinated intervention would be difficult to attain. When pollution becomes wicked, the structural and communicative complexities determine that no shared visions on the exact nature, scope and scale of the problem, nor a definitive, stable or well-defined solution can be held. Under such circumstances, problem-solving is often impossible due to the pragmatic reasons where deadlines are met or dictated by resource constraints rather than as the result of "correct" solutions identified. To pursue "solving" or "fixing" approaches may lead policymakers to act on unwarranted, unsafe assumptions and create unrealistic expectations. Therefore, addressing pollution usually needs a range of coordinated and interrelated responses, involving trade-offs of conflicting goals. Moreover, small changes of different regions can unfold largely unforeseeable differentia of system dynamics, leaving "traces" and creating path dependencies with "no right to be wrong", and no ultimately correct answers. The more wicked the problem is, the more likely a single intervention leads to the irreversible consequences. In this sense, intervention ambiguity means that effective interventions should be customized for different regions.

2.2.2. Pollution Wickedness of Social Accounts

Due to its ingrained boundary-spanning nature, more wicked pollution tends to generate conflicts of interests among multiple stakeholders when they attempt to frame and analyze the problems

according to their own perceptions, needs and interests. These conflicts themselves often create misleading frames that complicate matters even more when pollution stakeholders think of pollution issues in logical or illogical, and/or multi-valued thinking ways [28,35], from different contexts of state, market, civil society, and the leadership of organization. A typical example is that, the federal government of America has presented a big recession in the greenhouse problem, tending to deregulation, whereas some stakeholders in developing countries are more likely to view such pollution abasement as an important sustainable behavior [36].

How can the actions of stakeholders be inconsistent? We illustrate the details with the help of the societal triangle model. There are three important societal stakeholders or—in institutional terms—societal sectors surrounding and defining the problems, namely governments, firms, and civil society. The further pollution is beyond the grasp of the primary responsibility and core capabilities of each sector, the more wicked it becomes to come to effective solution-oriented approaches. The most-wicked problems are positioned in the societal centre, where the institutional void and the trust gap is the biggest. (Figure 2)

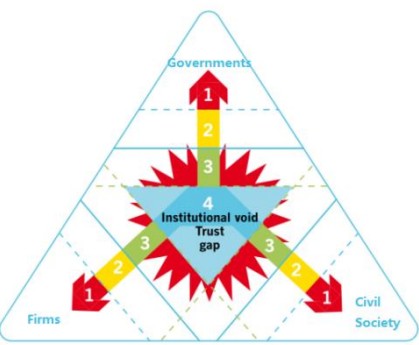

**Figure 2.** The societal intensity of wickedness [37].

1+1+1 Sectoral failure. Being exposed to the pollution, the sectors fails in their capability to produce sufficient goods and values, even when this is their primary responsibility. The market failure exists in the case that firms do not supply clean goods that people want or can afford—governance failure exists in the case that governments are incompetent with respect to pollution supervision and policy implementation, and civil failure exists in the case that communities do not lead environment-friendly lives.

2+2+2 Irresponsibility for handling negative externalities. When a sector produces a negative effect on society, costs to the system increase as a result. Theoretically, it is the trouble maker's responsibility to take care of the issue. But the fact is contrary. Take the industrial pollution as an example, if a company realizes that his or her share of the cost of the wastes they discharge into the environment is less than the cost for purifying the wastes, the company will be locked into a mindset of "fouling your nest", and tend to shrug off the environmental cost. Only when other sectors assign responsibility to them will the company be reactive to take it up. This could happen if governments regulate against pollution, or citizens and societal organizations protest against it.

3+3+3 Insufficiency of positive externalities. Some problems can be addressed by individual sectors, but it may run the risk of being underprovided if the problems are left to the initiating sector. This relates to the so called "merit goods", suggesting the commodity in which the society or an individual should have on the basis needs, rather than on his or her ability and willingness to pay. Furthermore, these active actions trying to fill in the void may run the risk of taking away the incentive for other sectors to contribute as well.

4+4+4 Systematical challenges. The so-called "common pool" problems are no-one's primary responsibility yet they affect everyone in the longer run. They are also referred to as "tragedy of the commons" and can be considered the most wicked in the social account. One instance is the increasingly serious plastic pollution today makes trouble to more and more coast countries though not all of them are the troublemakers. The existing governance system cannot address such global

environmental problems that have gone far beyond the boundaries of countries, industry sectors, consumers' habits, and human generations. Such systematical problems are also called "collective action" problems, as they require the joint collective actions of all societal sectors to be involved.

The variety of the societal sources of wickedness indicates that it may be difficult—not impossible—for a societal sector to take up responsibilities for the related issues that lie beyond his/her primary roles and capacities, even if the sector has long-termed interest in doing so. A well-functioning society should act as a "balanced" one, in which each societal sector plays constructive and complementary roles. The better each sector functions to be balanced, the more promising it will be in addressing the more wicked pollution situation.

### 2.3. Proposing Relationships between Wicked Pollution and Economic Growth

From above, we can see that each societal sector adds a different, complementary approach and logic to more wicked pollution, because the primary responsibility, main competencies, and main duties of each sector differ markedly from each other. To keep the problem-based societal ecosystem in a desirable shape, it is important that the opportunities should not mainly accrue to the "happy few" who are in the position to reap fruits, and it is also vital that opportunities do not mirror overly optimistic or superficial claims of the extent to which they are actually contributing to the resolution of the problem. So, if any societal sector actually takes an inactive attitude—as if to say "I don't have a personal desire, nor do I feel the social pressure to be engaged in pollution abasement"—the ecosystem will get corroded in the end, and inevitably a social mess will occur.

**Proposition 1.** *Wicked pollution will increase during the economic growth if any societal sector holds the inactive attitude, as shown in the first phase of the EKC.*

When increasingly serious situations trigger some outbreak of salient events, the responsive transition will be evoked, the situation can turn to "I will be engaged in pollution abasement, when I am reminded or called out by others, so that I can prevent penalties or negative opinions of others". The reaction of the government is strengthening supervision, and the reactive business leads to corporate technology improvement to minimize the harm [38]. However, the transition at most acts as a placebo because all the reactions also provide the evasions of their further action to fulfill primary responsibility. For example, the CEO of Exxon Mobil Corp., Rex Tillerson has claimed that there is no viable pathway for technology today to achieve the tipping point of the carbon dioxide level that is no harm to economies and well-being around the world [39].

So, any reaction can be seen as a window-dressing promise, refusing radical change to let in the intrinsic motivation. The same story could happen to other stakeholders, who are unable to skip the gaps to become active. In the case that pollution emissions cannot really be held under control, rising along with economic growth in the end, though temporary decline may occur due to technological progress, strengthening supervision or some other positive effects in the process.

**Proposition 2.** *The pollution growth curve presents a wavy line with an upward tendency when all the societal sectors take the reactive attitude—the deformation line of the first phase of EKC. (Figure 3, line 1)*

When one societal sector actively does too much beyond his/her "territory", but others remain reactive, it will lead to poor efficiency, and encourage others to reap the benefits as well. This may not be good news for sustainability. For example, if a corporate frontrunner does too much philanthropy in an environmental movement, some non-profit activities will be obviously deviated from its fiduciary duty. This is rather negative to its business and the economic sector. Similarly, if a government with over-active logic uses too many subsidies to encourage companies or non-profit organizations to take abatement measures, as a result, they will all suffer from a "subsidy addiction" which may negatively affect their capacity to stand on their own feet. In this situation, the entire economy would boom first, then bust in the end. For instance, the sales of China's plug-in electric vehicles, benefiting from

the massive central and local government subsidies, surged 343% in 2015 [40]. However, the China New Energy Vehicle Report (2017) reflects that the blind subsidies did not lead to a high amount of technology in the new energy vehicle sector, and statistics show that sales plunged after the reduction of subsidy at the end of 2018 [41]. In this regard, Dr. Winegarden states "If [the] government wants to encourage an electric car future, it should embrace the free market and remove the barriers to cheap and efficient car manufacturing that drive up costs too high for most drivers" [42].

**Proposition 3.** *When one societal sector holds the over-active attitude (far beyond its fiduciary duties) but others remain reactive, the pollution-growth curve will go downwards, but with an economic decline, indicating that environmental improvement occurs at the expense of economic growth (Figure 3, Line 2)*

It is unhealthy for an ecosystem when one societal sector is excessively active in address wicked pollution, it is also the same when two sectors are active to address pollution while the third remains reactive, because they would take over the fiduciary responsibilities of the rest. The effect is also known as "crowding out". The remaining sectors remain reactive because they want to avoid high costs and risk in the context of insufficient trust, or they are ready to be the free-riders to reap more from the ecosystem. For instance, when communities or governments clean up the waste produced by firms, they provide a perverse incentive for firms not to take the responsibility themselves. Anyway, it cannot be well addressed in such a situation, though there may be some temporary declines due to those active behaviors, showing an upward trend of wave.

**Proposition 4.** *When two societal sectors hold active attitudes towards more wicked pollution but the rest remain reactive, the pollution growth curve will present a wavy line with an upward tendency. (Figure 3, line 3)*

More wicked pollution requires the participation of all societal sectors at the same time, who however may not feel a responsibility, and may primarily see the risks of getting involved. Under this circumstance, societal attitudes towards pollution become resultant of rational choice for self-interests and societal uncertainty due to the lack of trust among societal sectors, often leading to the inconsistencies between the intention and the implementation, and the extreme difficulty in prioritizing as taking all trade-offs are taken into account. However, once pollution crises break out, the necessity of an immediate response is obvious, which is also called the situation of "inescapable wickedity" [43]. Only when all societal sectors hold the active attitude—such as to say that "I am motivated to be engaged in the abatement of pollution because it is part of my perceived identity and strongly held beliefs (I try to); I am also motivated to engage in sustainable activities on a regular basis (I really want to)"—can the wicked pollution get under control.

**Proposition 5.** *The pollution growth curve will no longer go up only when all societal sectors hold the active attitudes, realizing that pollution abasement is their own responsibilities.*

The resulting voids and transitional frictions not only generate wicked problems, from a different angle, the wickedness can be explored and leveraged as a means to drive breakthrough, and the development of new approaches to societal challenges, or "wicked opportunities" [44]. Diverse actors in the ecosystem can create new value through implementing productive and sophisticated models of collaboration [45]. The practical relevance of the idea of "collaborative advantage" [46] critically relies on appropriate cross-sector collaboration, embracing systemic goals and incremental and adaptive change, leaving no one out. A proactive attitude is highly demanded, such as to say that "I am engaged in pollution abatement as much as possible and encourage others to do the same in order to address the root cause of the issue". However, cross-sector collaborations with transformational power are not formed overnight because the ecosystem includes both collaboration and competition, and further, requires insightful and strategic considerations which only can be achieved with many trial-and-error practices. Thus, "wicked opportunity" comes with "collaborative complexity" [47].

**Proposition 6.** *Pollution will decrease with the economic growth when all societal sectors hold the proactive attitudes to take collective actions. Different from the second phase of EKC, however, the curve will present a wavy line with a downward tendency. (Figure 3, line 4)*

Under the right, powerful leadership, the wicked problems can be converted into opportunities more effectively. Since wicked problems are ambiguous, indeterminate, and boundary-less, and since enhancing ecosystem resilience calls for greater diversity and complexity, the heroic leadership is less likely to succeed than the open, inquiry-based, and collaborative style leadership that is capable to engage multiple stakeholders [48–50]. Different from the traditional kind of leadership that turns to "hard power" for command and authority with a focus on providing solutions, new leaders dealing with more wicked problems need to combine traditional "hard power" with "soft power", emphasizing the design and implementation of cooperative processes [48,51]. These leaders should be armed with the following characteristics: they should be able to think systematically, beyond the limitations of the organization or field, to form a positive common vision; be good at dealing with contradictions and conflicts; have a sense of social responsibility, and maintain trustworthy relationships in the process of cooperation [50]. A proactive organization needs proactive leadership to address wicked problems (see Table 1).

**Table 1.** The key dimensions of the leadership attitudes towards more wicked pollution.

| Key Dimensions | Four Types of Leadership | | | |
| --- | --- | --- | --- | --- |
| | Inactive | Reactive | Active | Proactive |
| Attitude towards pollution abasement | I cannot /negative | I have to | I can | We can |
| Attitude towards environmental protection | Is it possible/ profitable? | Do less harm | Do more good | Do better work |
| Response to the failure in addressing pollution | It is impossible/ non-profitable | Look for external causes | Look for internal causes | Seek for further cooperation |
| Action strategies and approaches | Transactional | Competitive | Heroic | Collaborative |

**Proposition 7.** *Proactive leadership is called for to encourage people and organizations to work together on more wicked pollution and ensure its decline along with economic growth.*

To address pollution, local efforts cannot be overlooked in the direction of sustainability. As fewer developed regions may face lower costs to address regional pollution due to the lower environmental impact at a lower level of development, Munasinghe (1995) proposes to develop an optimal growth path or a sustainable development "tunnel" through the EKC [52]. A similar path has been proposed to address this phenomena in China [53] and is probably underway in the ecological demonstration areas of China, which are often located far from the rich areas with small bodies of population, low income, and an abundance of natural resources. These regions could have very limited social resources to deal with their environmental issues, but these would not prevent them from obtaining effective environmental institutions and policies [54].

Since the Eco-Demonstration Construction Program was established in 1995, these communities have adopted sustainability approaches, where income growth and environmental protection are both the most important requirements. Very strict criteria, including more than 100 points, have been carried out to achieve sustainability, including ecological agriculture, soil and water conservation, the integrated use of crop residue and environmental education [55], promoting green economy like organic agriculture, tourism, and service industry. Furthermore, the independent evaluation mechanisms and the hotline for public supervision have been introduced to ensure the effective use of the special financial allocations. The eco-demonstration communities—most of which have a per capita income of less than one dollar per day (which is defined as extreme poverty by the World Bank)—have achieved both economic and ecological improvement, resulting in sustainable poverty alleviation.

The environmental improvement can be considered to be the result of effective organizing of societal actors rather than income growth [56]. Thus, it is possible to skip the first phase of the EKC and go straight to the second phase, and sustainability can be achieved in the early stages of economic growth.

**Proposition 8.** *Though the strong organization of societal actors, environmental and economic goals can both be achieved when the economy is at a lower level (Figure 3, Line 5).*

The environment is an inclusive term encompassing both natural and human systems. The latter refers to the interconnected aggregation of social governance systems [57]. In the context of China's centralized system, the governmental attitude plays an essential role in fighting for sustainability. A proactive governments can be a good organizer of multi-sector collaborations at the center of power of regional or national resources allocation. After all, sustainable development is at least as heterogeneous and complex as the diversity of human societies and natural ecosystems around the world [58], requiring an open, dynamic, and evolving mindset to fit in different situations through the cooperation and efforts of all societal sectors. Among them, Sagoff (1994) believes that moral and cultural values are central to both human experience and environmental policy [59]. Anyhow, we believe that the formation of a proactive attitude and mindset is not directly connected with economic growth.

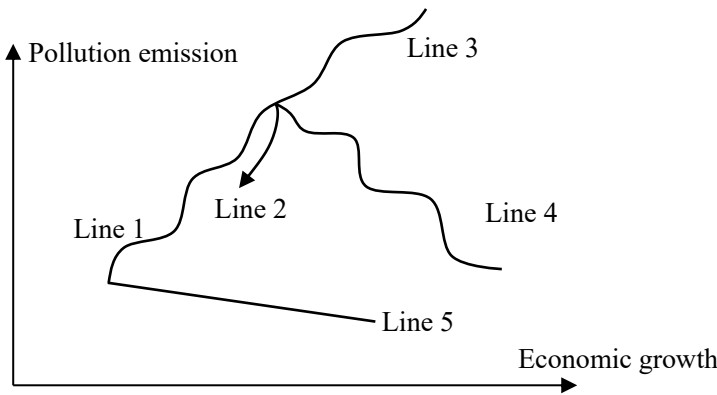

**Figure 3.** Possible relationships between more wicked pollution and economic growth.

As the proactive attitudes of societal sectors are needed to achieve the sustainability of interest, how will the proactive attitudes be achieved? Further, how will the attitudinal transitions be achieved? Next, against the societal background, we examine the formation process of proactive attitudes and the achievement roadmap of the attitudinal transitions.

*2.4. Attitudes of Societal Actors towards the More Wicked Pollution*

According to the EKC scenario, people's attitudes towards pollution change from the negative to positive, and their behaviors from the passive to active with economic growth [60]. It is our contention that the types of stakeholder attitudes towards more wicked pollution are determined by their motivations for sustainability from basic desires or from social norms.

Societal responsiveness can be intrinsic or extrinsic. The more people know what they want, the more they are able to frame strategies on the basis of intrinsic motives. Under this condition, people are often motivated by "I want … … ". If people do not know what they want or are primarily motivated by what they "don't want", extrinsic motivations prevail and social norms (must) take over. Social norms can be influenced by the culture of a country, but they are ultimately determined by the occupational group that people belong to. A very important argument in ethical reasoning is, for instance, to not do any harm, or to not do anything wrong, which can be regarded as the "negative duty". "I must" is the regular term used to assume liability. However, in terms of motivation,

it becomes clear that not doing harm could be important to establish basic social principles, but it does not provide any guidance in the way of "doing more good" for society, which can drive the operation of the whole social system more actively. The "I can" approach declares that people are willing to take social responsibility beyond fiduciary duty.

People's basic desire motivations are linked to social or group norms. If the group does not have significant influence, intrinsic motivations prevail. The attitude would be "I don't want" or "I want". Extrinsic motivations are linked to what people believe they "must do" or "can" then they can achieve. We illustrate the attitude model in Figure 4, which can further define the societal complexity of pollution.

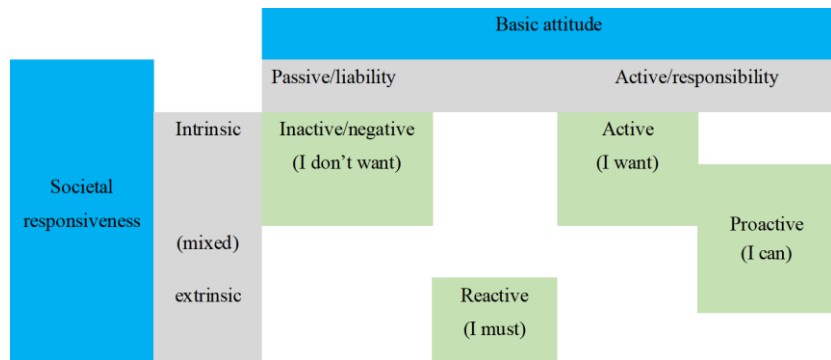

**Figure 4.** Attitude model and attitude types towards the more wicked problems [61].

The intrinsic motivation is a fundamentally introverted attitude based on the human utilitarian-oriented motive. When a societal actor find he or she is far away from the pollution, or it does not help his/her fiduciary duty or has no foreseeable interests to make pollution abasement, the actor will leave the majority of the responsibilities with other stakeholders, and, accordingly, takes an inactive attitude for the pollution. However when the actor learns, for instance, from the shocked reports and troubling pollution-generated results, that the situation is deteriorating, and beginning to threat his/her regular life and working places, the attitudinal transition will lead to two possible outcomes, forming the activation route or the responsive route (Figure 5). In the former case, the actor finds it is necessary to take responsibility with a more moral and strategic attitude, regardless of society's response, and the actor is intrinsically motivated to move forward and becomes a front-runner. In the latter case, the actor tends to take necessary actions according to the response of other pollution-related stakeholders when the actor is faced with serious life-threatening problems.

The extrinsic motivations are linked to what people want and what they can achieve. If people mainly do things that they have to do, they will become reactive and extrinsically motivated. There is an argument to interpret the trade-off: people and organizations facing sustainability issues does not necessarily make moral choice, and, and when they make the choice, it is not necessarily with strong willingness. They just do not have the ability to make the necessary changes due to the lack of information and resources. Interestingly enough, once the problem is addressed, the reactive actor will be active and eager to inform his/her closely related stakeholders of what he/she has done for good reputation. Once the limited information and resources benefit from an effective breakthrough, people and organizations could become intrinsic to address pollution, that is, they will find they are "capable" of doing more good. This change is called the capabilities route (Figure 5).

When societal sectors realize that the pollution should be addressed through cooperation and coordination of all stakeholders, they become to be a proactive actor, will take action proactively, and initiate strategic dialogues with all related stakeholders. Accordingly, the great difference between activeness and proactiveness is whether a set of strategic dialogues and serious cooperation are involved within stakeholders [62], which will combine the intrinsic activeness and the extrinsic reactiveness to other stakeholders' decisions of the alliance. The change is defined as the collaborative route (Figure 5).

It can be promising if, among all stakeholders, the consensus can be aroused and the strategic alliance can be formed to proactively address pollution before the ecosystem collapses.

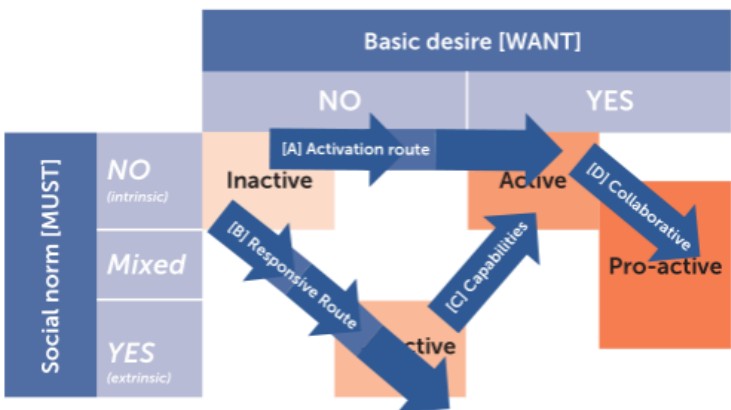

**Figure 5.** Four transition routes of attitude towards wicked pollution [61].

Since attitudes rely on the interaction of intrinsic and extrinsic motivations, any change of the interaction can arouse attitudinal transition. Certain triggering events can bring people out of their established routines as automatic cognitive processing occurs [63]. They can also act on the change of an original social norm beyond the intrinsic logic of societal actors, but the routes of attitudinal transitions are different accordingly. The triggering event is characterized as a complex of three interacting components: high strength (its novelty, disruption, and criticality), broad space (the effects can spread through the whole organization or society), and the long-span time (when the event occurs, it remains impactful for a long period of time) [64]. Moreover, as illustrated above, the "effective" event should focus more on danger, looking at what needs to be done now than what can be done in the future due to knowledge framing towards the wicked problems. Eventually, the collaborative route of attitudinal transition should be achieved to form collective actions among societal sectors.

## 3. Results and Discussions

In this paper, through an extensive literature review, we show that the extant EKC pattern is only valid for the pollution of a less wicked nature, since such pollution can be dealt with by technical progress and rational actions during the period of economic growth, whereas the more wicked pollution is full of ambiguities and societal uncertainties. Based on the theoretical and empirical literature review, we propose a set of propositions on the relationships with the economic growth, showing potential trajectories which can be very different from the conventional recommendations of the EKC.

In particular, our Proposition 6 claims that involved societal sectors should hold the proactive attitudes to address such pollution through the cross-sector collaboration. The key missions cannot be achieved without the collaborative attitudinal transition, in which the triggering events related to pollution could promote attitudinal transitions. Both Propositions 7 and 8 indicate that proactive leadership and the strong organization of societal actors can ensure the effective collaboration, which could be achieved even at a lower level of the economic growth.

### 3.1. Contributions to EKC Study

This study makes contributions in developing a conceptual base framework to understand the relationships between more wicked pollution and economic growth, which should blend the rational accounts of optimal decision-making with the social accounts of conformity pressure. Societal actors are assumed to be self-interested, norm-free, and opportunistic maximizers of short-term interest, who typically neglect the long-term interests of the collectivity [65], which is not valid for more wicked pollution.

On the one hand, such pollution is full of ambiguities, which greatly prevent people from getting optimal solution. One the other hand, the social accounts emphasize that commons are not predetermined but socially constructed phenomena, in which social norms motivate actors' behaviors instead of individualized rationality [66]. Collective awareness of the complicity in creating a commons problem heightens actors' collective identification with the problem and allows for practices of collaborative engagement to remedy the problems [67,68]. Based on the notion, we argue that actors' attitudes towards more wicked pollution is the result of social norms—a link between basic desire and social responsiveness. The attitude classification of this study can illustrate the evolutionary process that wicked problems emerge from the individual rationality, become wicked under social pressure, and finally can transform into wicked opportunities, when well addressed by collective mitigation measures based on social collaborations and efforts.

The prior EKC literature assumes a homogeneous attitude of stakeholders to address the pollution problems from the negative to positive as the particular situation gets worse, and then the desirable economic growth, along with the declined emissions. We argue that the wickedness of pollution cannot be perceived and dealt with by different stakeholders in the same manner. The perceptions and mindsets of the stakeholders can be quite different across three main societal sectors. Hence, the attitudes towards the environmental issues are probably asynchronous, from inactive, to reactive, to active, and to proactive. The attitudinal combination can lead to various patterns of the pollution-growth curve. If the attitudinal transitions are taken into account, the curve would assume multi-stage characteristics and greater diversity. Furthermore, it is our speculation that the inverted U of the EKC could be a section of the whole curve, so is the N-shaped or the inverted-L or the cubic shaped curve in the extant EKC empirical study.

### 3.2. Contributions to Wicked Problems Theory

Our work also has several important implications for research on wicked problems. First, we extend the work to elaborate on the societal intensity of wickedness, showing how the solutions are beyond the primary responsibility. Rittel and Webber and followers have depicted wicked problem as various intractable, interdependent, and socio-economic issues, in particular requiring other manners of diagnosis and thinking [69]. We argue that pollution problems can take various degrees of wickedness, requiring discrepant efforts to address: the less wicked may need technical progress and rational decisions, which can be addressed along with the economic growth, whereas the more wicked need collaborative actions, proactive leadership and strong organizing of societal actors, which holds no direct relation with the economic growth.

Second, focusing on the environmental issues, we recognize that social accounts can play important roles and thus we propose the dynamic process of people's motivations in addressing more wicked pollution. Ansari et al. (2013) shows that five mechanisms, including collective theorizing, issue linkage, active learning, legitimacy seeking, and catalytic amplification, underpin how and why actors change their frames toward greater consensus around a common logic [68]. Our results are partially consistent with the findings. We argue that salient events, acting as the catalyst, can promote collective theorizing, issue linkage, active learning, legitimacy seeking. Meanwhile, we believe that proactive leadership and strong organizing of societal actors are particularly required in the process. Moreover, Lazarus (2017) believed that the purpose of environmental strategies is to protect the future at the expense of the present [30]. Instead, our study argues that the wicked problems can be converted into new opportunities with proactive attitudes, not necessarily at the expense of the present, for example, unexpected niches and new fields for social development can arise afterwards. Therefore, wickedness can be explored and leveraged as a means to drive breakthroughs.

Turning to the case of the construction of ecological demonstration areas in China, where the effective environmental governance has been made, we agree that a partial or a higher level of concerted actions can be reached through coercion rather than negotiation due to the more stakeholder power in the local or national contexts [68]. However, it is important to mention that the keynotes to address

wicked problems are the formation of a commonly agreed upon design and a good collaboration mechanism, rather than the prevailing power, though it could be helpful to formulate regulation and enforce it.

*3.3. Methodological Implications*

As many of our methodological prescriptions and techniques have been developed for attitude-and-event-oriented research, one key issue revolves around the measurement and research design of the research itself. When a research focuses on qualitative descriptions of organizational attitudes, records, reports, a research design focusing on behavioral observations may be appropriate. When it is qualitative interpretations and/or reactions to the events of interest, the researcher may better methodologically consider interviews, observations, and questionnaires. Meanwhile, the advent of social media (e.g., blogs, Facebook, Twitter) and the increasing ubiquity of audio and video monitoring of the public places provide many other methodological opportunities. Furthermore, we should take the prevailing attitude and the attitudinal transition process into account when conducting societal-oriented research. Many process-oriented scholars offer a range of potential ways to use qualitative methods to analyze associated issues and their effects on organizational practices and change [70].

**4. Conclusions**

Ultimately, the environmental issues are wicked problems at various degrees. The less wicked pollution—as it is more technical and regional—can be gradually resolved since the economy provides appropriate technical resources, and law establishment and enforcement, most of which fit perfectly for the EKC hypothesis. In contrast, the more wicked problems of a greater societal, transboundary, and transgenerational nature have been most challenging. At best, those problems can only be resolved and may surface over and over again for the human bounded rationality and societal uncertainty in due course. If organizations and practitioners tend to address such pollution with conventional mindsets—optimal or moral—it will come to a wicked mess. Under these circumstances, the relationships between economic growth and pollution can be more uncertain, and will not be in an inverted-U shape.

The wicked problem can also be reframed as a wicked opportunity under the condition that all societal sectors exert their unique functions to take collective actions on the willing, not on the need. Governments, firms, and civil societies are three important societal sectors defining the pollution, whose attitudes and behaviors critically determine the pollution growth trajectories. The relationships shall be very diversified, e.g., a synchronous line, a wave line with the upward tendency, or a wave line with the downward tendency, given the three societal sectors take different attitudinal portfolios. Among them, the proactive attitude is necessary to form cooperation and coordination among all stakeholders to realize both environmental and economic goals. Moreover, the well-structured governance of such collaboration requires proactive leaders and strong coordination of the societal sectors. Achieving a proactive attitude is a prerequisite task in addressing such pollution. There are four routes of attitudinal transition driven by the salient events, and the collaborative route of attitudinal transition can be achieved to establish a set of strategic dialogues and cooperation among stakeholders.

Integrating and synthesizing the theories and empirical findings of the extant EKC literature, this paper proposes a conceptual base framework and theoretically demonstrates that different patterns of pollution emission can be attributed to the pollution of various wicked degrees, and different attitudes of societal actors towards the more wicked pollution. This theoretical clarification contributes to the research of wicked problems in that the problems of wicked degrees lead to different causal relationships with economic growth that requires different strategies, leadership, and societal efforts to address. In doing so, we hope to advance the conversation on EKC studies and the abatement practice adaption.

However, for empirically pursuing a wicked-oriented program of research, a range of methodological tools and data analysis issues should be taken into consideration. Furthermore, as to the catalyzers in attitudinal transitions, some triggering events may arise without deliberate planning, and others can be created strategically to achieve the desired effects. Thus, exploring research in this avenue can be of great value to policy programming. In addition, contemporary partnership practices have been criticized for not adequately addressing systematical changes. In this regard, how to address the "collaborative complexity" is an important avenue for future research.

**Author Contributions:** Conceptualization and methodology: J.C.; English editing: T.E.H.; writing-original draft: J.C. and R.v.T.; writing-review and editing: J.C. and T.E.H.; funding acquisition: J.C.; project administration: J.C. and T.E.H.

**Funding:** This research was funded by Jiangsu Social Science Project (18SHD001), and the Key Projects of Social Sciences of Jiangsu Education Department (2018SJZDI083).

**Acknowledgments:** The authors would like to sincerely thank the three anonymous reviewers for their insightful criticism, encouragement and review comments on our manuscript. And, the authors would like to thank the Associate Editor and Editor-in-Chief for the research opportunity and their encouragements in the process of revision and resubmission.

**Conflicts of Interest:** The authors declare no conflicts of interests.

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
