# Peer review of "Is the Environmental Kuznets Curve Still Valid: A Perspective of Wicked Problems"

_sustainability, doi:10.3390/su11174747_

Round 1

Reviewer 1 Report

The quality of the article is high because it deals with a rather “independent” analysis of the EKC in a fluctuant context (that of “wicked problems”) focusing on wealth perform and attitudinal processes towards environmental damage. The work wishes to enhance the gap triggered by the conflicting reactions arisen through the testing of the model EKC in various manners by introducing the theory of wicked problems to reframe the extant EKC research into a common context that of pollution (as expressed in the abstract)

As it is known, there is no definitive formulation of wicked problems, because (i) they have no stopping rule; (ii) solutions to wicked problems are not “true or false”, but rather “good or bad”, and there is no immediate and no ultimate test of a solution to a wicked problem; (iii) every solution to a wicked problem is a “one-shot operation” and wicked problems do not have an enumerable (or an exhaustively describable) set of potential solutions, nor is there a well-described set of permissible operations that may be incorporated into the plan; (iv) every wicked problem is essentially unique, and every wicked problem can be considered to be a symptom of another problem (P. Thollander, MDPI, 2019).

A little forward the authors will enlighten this fact especially because it is clear that most of environmental issues involve multiple interacting systems, replete with not only rational choices but also social and institutional uncertainties (lines 93-96). The authors follow to enlist the pros and cons of EKC:

EKC and its pros for environmental protection (lines 27-64):          

-          Panayotou (1993) finds a strong link between the pollution emission and the economic growth in an inverted-U curve;

-          the industry-upgrading model proposes the transformation mode of the economic development from the polluting industrial economy to the clean service economy (e.g., Panayotou, 1993; Arrow et al.,1995);

-          green Solow model claims the process of emission reduction arising from the exogenous technology progress, leading to the greater usage efficiency in energy and materials;

-          the displacement model (Antweiler et al.,2001) introduces a theory believing that the higher environmental standards in developed areas push or outsource the dirty industries to the underdeveloped areas.

EKC and the cons when the research on the cause of pollution extends to the social system and national context (lines 65-79):                                                                                                                    

-          Kaika and Zervas (2011) finds that the EKC place too much emphasis on the production, arguing that the improvement of technological structure would be offset when the final consumption holds the pollution intensive, especially under the constant growth of urban wastes (Wagner, 2010);

-          several other studies (e.g., Panauotou, 2000; Dinda, 2004; Lieb, 2004) point out that most of empirical studies have assessed the turning point of the EKC based on the average income level of the countries, assuming that the world income is normally distributed – yet the world distribution is indeed highly skewed (Milanovic, 2002; Roser, 2015), thus it is rather unrealistic to turn to the average income to estimate the turning point of the EKC;

-          Grimes and Roberts (1997) state that the EKC patterns are valid only for those developed countries with a colonial history, geopolitical powers and trade superiority. Those patterns may not be as valid to the less developed world of today (Nahman and Antrobus, 2005), implying that the EKC pattern may not be repeatable for developing countries.

Please check the Panyotou surname at line 70 or better transform it with a no. as for MDPI style.

I mainly found your interesting manuscript could be improved through a better management of paragraphs, as often performed by the following order: Introduction, Materials /Methods, Analysis, Discussion.

To this end:

1)       I would re-organize the paper with few modifications to the text, but a robust restructuring of sequences of your thoughts and related comments

2)       I would add some information more about what are the main problems of pollution: example, supervision, release, process systems.

3)       I would stop at line 104 the “Introduction” and open a new paragraph: provisional title “1.1. The background scenario for reframing the extant EKC into an attitudinal mind-set towards pollution

4)       At lines 151-152, I would add some other example of wicked problems.

5)       I would move your paragraph (original lines 176-228), after 2.1. Concept and nature of pollution wickedness. Therefore, starting at line 153 – renumber it 2.2. accordingly.

6)       Then, I would open as a new paragraph your explaining of why pollution is a “super-wicked problem”: provisional title “2.3. Pollution in-between the lacking fulfillment by the authorities and the call-to-action”.

Here (original line 229) you can therefore start to show the ambiguity created by the structural complexity of pollution through the knowledge, predictive, and intervention ambiguity you righty pointed out, and then (original line 289) you enter your own methods “Proposing relationships of wicked pollution and economic growth analysis” (original lines 348-488) and terminate with your own analysis “Attitudes of societal actors dealing with wicked pollution” about the transition of societal attitudes towards pollution (original lines 289-347).

Figure 2 is more a table (lines 336-337). By the way, I would add “Negative: I can’t”.

The same in your table 1 (lines 450-451), eventually maintain inactive adding “/negative

YOUR Propositions 8: The environment can be under effective governance through the strong organization of societal actors along with a low level of income (lines 476-477).

7)       Finally, to improve this assertion, I would add an annotation on the meaning of the expression “strong organization of societal actors,” making use of Mark Sagoff’s Four Dogmas (1994). But this it’s just an advice, any of compulsory.

Lines 556-558: it is nice to read about “Acknowledgments”, but the character is too big. Check the ch. points.

I noticed at line 685 there is the MDPI licence with the year 2017    by    the    authors.    Submitted    for    possible    open    access    publication    under    the     terms and conditions of the Creative Commons Attribution (CC-BY) license (http://creativecommons.org/licenses/by/4.0/). Check the template you use.

The paper is anyway well formatted and there is only to adopt the MDPI style for references infra-text.

With Kind Regards,

Reviewer 2 Report

The submitted manuscript is a new approach to the discussion on the environmental Kuznets curve hypothesis. There are serious flows in the paper preparation, which should be addressed.

The paper is not well-formatted. Please follow Authors Instructions to re-arrange the manuscript, Sections Introduction should be without sub-title. Sections Material and Methods, Results mandatory. The Discussion is not sufficient and should be extended. Conclusions should be added with explanation of the main scientific results obtained by authors. In the present form the article is not easy to follow – it should be pointed out the main ideas proposed, with more emphasize on the research novelty and practical applicability. 

Reviewer 3 Report

Authors present results of works on „Is the environmental Kuznets curve still valid: A perspective of wicked problems”.

In my opinion the topic undertaken in the paper has a long research tradition and is worth of scholars’ attention. However, the manuscript is burdened with numerous flaws and does not meet the high publishing standards of the Sustainability journal. This leads me to a conclusion that it cannot be accepted for publication in the journal.

Below I will enumerate some of the problems which needs to be tackled by the authors in their future research efforts:

-          The abstract of the paper is chaotic, misleading and does not introduce well to the content of the paper:

o   It suggests that it offers a “theory of wicked problems” – well, after reading the manuscript I cannot see any theory developed; rather, the paper attempts to offer some kind of conceptual framework but definitely not a theory,

o   It remains unclear what is the actual purpose of the paper – authors argue that the develop: a theory of wicked problems, then “a framework that, that facilitates the assessment of pollution wickedness, trace the source of the wickedness, and demonstrate the underlying evolution and transition of societal attitudes” and finally a set of propositions “surrounding the relationships between wicked pollution and economic growth”; In my opinion this is too ambitious program for a 16-page long paper which at the end fails to provide any of these goals,

o   “controversies are still controversial” – please proof read you next papers.

-          The introduction is quite long and introduces many angles from which the EKC has been investigated. However, it is hard to grasp how does the intro links with the purpose of the paper. For instance, in the 2nd paragraph on the 3rd page you draw attention to the methodological problems related to analyzing the EKZ problems (omit variable bias, model adequacy). I can hardly see any connection with the further reading of the paper.

-          I think it is worth to introduce the reader to the literature applying the EKC concept to the wicked environmental problems (if there are such studies).

-          In the section 2.1 you attempt to introduce the concept and nature of pollution wickedness. You cite a study by Rittel and Webber but frankly you do not define what do you mean by wicked problems. It remains unclear to what purpose the subsequent paragraphs (in this section) serve. As a reader I cannot find a clear plot in these paragraphs. Neither can I link this section and the paragraphs with the subsequent sections of the paper. Please, be very clear and precise when writing such section as it is rather important for the whole manuscript.

-          The section 2.2 is devoted to the assessment of pollution wickedness. In this section you name five criteria that, in your opinion allow to assess the “complexity of pollution”. Do you then assess the pollution wickedness or pollution complexity? Or are the two synonyms? Based on what did you come up with these 5 criteria? Is it based on any literature? Nevertheless, these five criteria are not well defined, they seem no to be mutually exclusive and most importantly I cannot see how do you apply them in the remaining of your manuscript. Moreover, it could be argued that the these five criteria are source of pollution wickedness (section 2.3) while the three sources (of pollution wickedness) that you name in the section 2.3 (knowledge ambiguity, predictive ambiguity, intervention ambiguity) are actually the sources of problems when defining and measuring the wickedness of the environmental problems. In short, the content of the 2.2 section seems to be better fitting to the section 2.3., and vice versa.

-          Most importantly, it is difficult for a reader to understand how do sections 2.2 and 2.3 relate to the purpose of the paper.

-          In the section 3 you introduce the concept of societal actors. On the previous 7 pages you did not signal to the reader that your paper will circulate around this topic and that you will use is to confront the EKC with wicked problems. Then you limit the societal actors to the government, firm and citizens.

-          You cannot treat a market as a social actor. Also, you cannot treat communities as citizens, and vice versa.

-          I do not understand your statement at page 7 that “communities provide the social goods for communities”.

-          You devote a separate subsection to the matter of evolution and transition of societal attitudes. However, you do not link these attitudes with the purpose of the study. Also, you do not present the evolution and transition of these attitudes. In addition, you mix various terms such as societal attitudes, societal responsiveness, motivations. You do not define these  terms and do not show the relationships between them.

-          Most boxes in the table 2 are empty.

-          The propositions you offered are difficult to understand and not well supported by arguments. Some of them are tautological, i.e. you cannot reject them empirically (e.g. proposition 1). It is hard to develop or explain a proposition in one or two paragraphs and this is the case in this manuscript.

-          In the discussion section you do not refer to the topic of your paper but rather you introduce new terms such as collective awareness.

-          There are numerous problems related to the quality of the English writing. E.g.: Page 7 “Some pollution is not wicked such as “flowing water purifies itself every 10 miles”, “fouling our own nest” (Hardin, 1968), and the corresponding EKC is “clean waste later” so long as people behave only as independent and rational decision-makers.” – What does it mean?

-          Some literature cited in the text is not cited in the reference at the end of the paper e.g.: Huxham and Vangen, 2004; Teerlak and Gong, 2008.

Minor mistakes:

2nd affiliation has a mistake in the name of Uni.

Abstract: change “heavy” to “a lot of”, make it “Environmental Kuznets Curve”, Authors use term “wicked” too often.

Line 50: “are linked with”

Line 55: what do Authors understand by “seminal” here? Casing future developments?

Line 67: “find”

Line 69: how can “urban wastes” grow? It is too trivial statement, volume of generated wastes may grow.

80-89: this entire paragraph needs rephrasing; pollutants are not long-term – they are persistant, of chronic action.

Line 103: “in” not “In”

199: “include”

214: please explain what do Authors understand by “stakeholders”?

222: remove “,”              

Round 2

Reviewer 2 Report

I appreciate the work done by Authors, however, serious additional revision is still required. The format is not re-arranged in accordance with Authors Instructions, the detailed description of the new scientific results are not presented in the section Results, which is too small for the research paper. Some specific comments:

1.           Sections 1-3 should be united in a single section Introduction.

2.           Please re-arrange your sections Methods and analysis and Results into section Materials and Methods and Results. The section Materials and Methods should be reduced, while the corresponding text should be a part of Results section. Thus, you should make a distinction betwee applied methods and results of your study.

3.           Check the size font size at lines 180-196.

4.           Check text at lines 268-269.

5.           Please avoid the subscript references and re-arrange your reference list, by inclusion all these references.

6.           Re-name your section Conclusions and Further research as Conclusions.

7.           Add the Authors' Contribution information.

Reviewer 3 Report

Authors have increased quality of work and it can be accepted.

Author Response

We are highly appreciative for criticism, encouragement and insightful comments and suggestions that you have made on our manuscript in the first round. We have thought over each of the issues carefully, searched additional literature for answers, and performed significant rework on the manuscript based on our best efforts to meet your expectations.

We very much appreciate the chance to refine the paper so that our research shall find its way to publication in the journal, Sustainability

Round 3

Reviewer 2 Report

Some amedments have been introduced, however, the manuscrit needs further deep revision. The major problem in my opinion is that the type of submission "Article" is not appropriate to this kind of manuscriot.

Introduction should be presented without sub-sections.

Materials and Methods should be re-written. In current form this section doesn't give a clear presentation of the methodology.

Section Results provides a set of propositions and Authors ideas, without strong substantiation.

Discussion and Conclusions are too short.

Format needs further improvement (e.g. Figures captions).

Taking into account that this is a third round of review, without a required progress - I recommend to reject the current version, but encourage re-submission, as different type of manuscript (e.g. Viewpoint, or other).

Round 4

Reviewer 2 Report

The cover letter written by Authors to Reviewer is rather emotional, and I’m sorry that according to the principals of the scientific objectivity still cannot approve the publication without further revision.
First of all, my advice is to study again the Authors’ Instruction. You can find out after that - only a section “Conclusions” is optional, while all other sections listed are mandatory. Section Results however can be combined with section Discussion.
Lines 123-127 repeat the text from lines 109-113.
In section Materials and Methods “New methods and protocols should be described in detail while well-established methods can be briefly described and appropriately cited” (Authors’ Unstruction). Please follow this recommendation – in the present version of your paper this section doesn’t correspond to the journal requirements.
Your arguments regarding the critical literature review, and revised title of your manuscript reveal, that the paper type should be “Review”. This confirms my doubt in the correctness of the type of your submission. If you are sure, that reviewer should consider your paper not as a research article, but as a review – the manuscript should be re-submitted, and considered with different criteria, than the current submission.